# Metal-catalyst-free gas-phase synthesis of long-chain hydrocarbons

Lidia Martínez [1,17], Pablo Merino[1,13,17], Gonzalo Santoro [1,17], José I. Martínez [1], Stergios Katsanoulis [2], Jesse Ault[3], Álvaro Mayoral [4,5,6], Luis Vázquez [1], Mario Accolla[1,7], Alexandre Dazzi[8], Jeremie Mathurin[8], Ferenc Borondics [9], Enrique Blázquez-Blázquez [10], Nitzan Shauloff[11], Rosa Lebrón-Aguilar [12], Jesús E. Quintanilla-López [12], Raz Jelinek [11], José Cernicharo [13], Howard A. Stone [14], Victor A. de la Peña O'Shea [15], Pedro L. de Andres[1✉], George Haller [2✉], Gary J. Ellis [10✉] & José A. Martín-Gago [1,16✉]

Development of sustainable processes for hydrocarbons synthesis is a fundamental challenge in chemistry since these are of unquestionable importance for the production of many essential synthetic chemicals, materials and carbon-based fuels. Current industrial processes rely on non-abundant metal catalysts, temperatures of hundreds of Celsius and pressures of tens of bars. We propose an alternative gas phase process under mild reaction conditions using only atomic carbon, molecular hydrogen and an inert carrier gas. We demonstrate that the presence of $CH_2$ and H radicals leads to efficient C-C chain growth, producing micron-length fibres of unbranched alkanes with an average length distribution between $C_{23}$-$C_{33}$. Ab-initio calculations uncover a thermodynamically favourable methylene coupling process on the surface of carbonaceous nanoparticles, which is kinematically facilitated by a trap-and-release mechanism of the reactants and nanoparticles that is confirmed by a steady incompressible flow simulation. This work could lead to future alternative sustainable synthetic routes to critical alkane-based chemicals or fuels.

[1] ESISNA group. Instituto de Ciencia de Materiales de Madrid (ICMM-CSIC), c/ Sor Juana Inés de la Cruz 3, 28049 Madrid, Spain. [2] Institute for Mechanical Systems, ETH Zurich, Leonhardstrasse 21, 8092 Zurich, Switzerland. [3] School of Engineering, Brown University, Providence, RI 02912, USA. [4] Instituto de Nanociencia y Materiales de Aragon (INMA), Spanish National Research Council (CSIC), University of Zaragoza, 50009 Zaragoza, Spain. [5] Laboratorio de Microscopias Avanzadas (LMA), University of Zaragoza, 50009 Zaragoza, Spain. [6] Center for High-Resolution Electron Microscopy (ChEM), School of Physical Science and Technology, ShanghaiTech University, 393 Middle Huaxia Road, Pudong, Shanghai 201210, China. [7] Catania Astrophysical Observatory (INAF), Via Santa Sofia, 78, 95123 Catania, Italy. [8] Institute of Chemical Physics, Université Paris-Saclay, 91400 Orsay, France. [9] Synchrotron Soleil, L'Orme des Merisiers, Saint-Aubin - BP 48, 91192 Gif-sur-Yvette, France. [10] Instituto de Ciencia y Tecnología de Polímeros, ICTP-CSIC, c/ Juan de la Cierva 3, 28006 Madrid, Spain. [11] Department of Chemistry, Ben Gurion University of the Negev, Beer Sheva 84105, Israel. [12] Instituto de Química-Física Rocasolano (IQFR-CSIC), c/ Serrano 119, 28006 Madrid, Spain. [13] Instituto de Física Fundamental (IFF-CSIC), c/ Serrano 123, 28006 Madrid, Spain. [14] Department of Mechanical and Aerospace Engineering, Princeton University, Princeton, NJ 08544, USA. [15] Photoactivated Processes Unit IMDEA Energía, Av. Ramón de la Sagra, 3 28935 Móstoles, Spain. [16] Institute of Physics of the CAS, Cukrovarnicka 10, Prague, Czech Republic. [17]These authors contributed equally: Lidia Martínez, Pablo Merino, Gonzalo Santoro. ✉email: pedro.deandres@csic.es; georgehaller@ethz.ch; gary@ictp.csic.es; gago@icmm.csic.es

Carbon–carbon (C–C)-bond formation is at the root of organic chemistry and is a fundamental building block for many essential materials in modern society, from pharmaceuticals to polymers. In particular, alkanes are important raw materials for the chemical industry and form the main constituent of fuels and lubricating oils, their predominant sources being natural gas and crude oil. In recent years, extensive efforts have been devoted to developing sustainable methods for the generation of alkanes and their derivatives that impact many scientific disciplines, including chemistry, materials science, energy, and health. Many technologies have been established, including transition metal–catalysed processes such as olefin metathesis and cross-coupling[1–3] or the production of clean fuels and chemical feedstocks via advanced catalytic approaches in, for example, Fischer–Tropsch synthesis (FTS)[4–6]. FTS is a mature technology reaching an average conversion of 45–50% and in some specific cases as high as 90%[4,7]. However, the control of selectivity to long-chain hydrocarbons is highly challenging and cannot exceed 45%, as predicted by the Anderson–Schulz–Flory (ASF) law[5,8]. It is also important to highlight the recent developments in alkane production via microbial biorefineries[6], and photocatalytic production of hydrocarbon from fatty acids[9], but these are still at low-technology development levels. It is clear that all of these processes play vital roles in the drive toward cleaner and energy-efficient technologies for future sustainable chemicals and fuels that are critical to counteract the harmful effects of global warming and fulfil future societal needs[1,10]. However, at the present time, most industrial processes leading to C–C chains inevitably depend on expensive or critical raw materials, such as metal catalysts, non-environmentally friendly or toxic solvents, or require harsh reaction conditions in terms of temperature or pressure. Thus, there is certainly room for new approaches that can lead to groundbreaking technologies to provide for our future materials and energy needs.

Here, we describe an alternative route to the generation of long hydrocarbon chains under radically different conditions to current approaches, employing carbon atoms and molecular hydrogen as precursors in a gas-phase process at moderate temperature (<450 K) and low pressure (<0.5 bars) without the use of metal catalysts. While alkane formation under these conditions is theoretically unexpected, we demostrate experimentally and we rationalise the results using theoretical calculations to uncover the important role played by $CH_2$ and H radical and carbon nanoparticles (C–NP) in the overall process showing that, albeit kinetically unfavourable, C–C chain growth is energetically possible on the surface of carbon nanoparticles. This process is schematically represented in Fig. 1a.

## Results

**Synthesis of hydrocarbon chains**. The synthesis is achieved by using a sputter gas-aggregation source[11] (see supplementary Note 1), especially developed to produce gas-phase C clusters and C-NPs[12]. In this system, the sputtering gas (Argon) flow also serves as dragging gas. The careful design of the reactor conditions facilitates, via a fluid dynamics-based trapping and release mechanism, a large increase in the residence time of the reacting species, which leads to the efficient formation of long hydrocarbon chains with a significant reaction yield. These chains are entangled to form long micron-length fibres. Figure 1b shows a mat of fibres collected on a silicon oxide surface during around 100 min, with a conversion >50% and a $C_{24+}$ selectivity >70% (see more details about efficiency of the process and scaling possibilities in the Supplementary Note 2) with an average density estimated from the images of about $10^{10}$ fibres/cm². Images obtained from single fibres show that they can reach up to 3 μm

in length and exhibit diameters between 6 and 20 nm (Fig. 1b, c). More detailed HR-TEM images show atomic features of 2.2 Å with an intermolecular spacing of 3.6 Å (Fig. 1d, e, Supplementary Note 2 and Supplementary Fig. 2), which are compatible with semicrystalline alkane structures (see also Supplementary Note 6)[13,14]. Interestingly, when no hydrogen is used in the process, we form only amorphous C-NPs (see supplementary Note 2)[15]. Some of these are visible in Fig. 1c.

Real-time mass spectrometry analysis of the reaction products in the gas phase shows that they are mainly composed of short aliphatic molecules in the $C_1$–$C_4$ range (Fig. 2a), although the electron-impact dissociation pattern prevents quantification of these species. Gas chromatography–mass spectrometry (GC–MS) of the fibres reveals that they are essentially composed of unbranched alkanes, with a relatively narrow distribution of $C_{23}$–$C_{33}$ (Fig. 2b), with a probability chain growth close to unity according to an ASF analysis[6]. Raman spectroscopy (Fig. 2c) confirms the alkane structure of the fibres (see Supplementary Note 6), revealing characteristic spectral fingerprints of solid-state semicrystalline long-chain n-alkanes[16–18]. Further, AFM-coupled infra-red photothermal nano-IR spectra recorded from an individual fibre (spatial resolution is 10 nm, Fig. 2d) clearly show methylene and methyl vibrational deformation modes (Fig. 2e) comparable to those obtained from n-alkanes with intermediate chain lengths in the $C_{23}$–$C_{33}$ range (see Supplementary Notes 6 and 7).

In our approach, gas-phase reaction occurs to produce alkane chains and fibres, avoiding surface diffusion and reactor-wall adsorption, usually occurring in arc-discharge or physical–chemical vapour-deposition techniques[19–21]. Here, in contrast to other procedures, the long alkane chains and subsequent fibres develop in the gas phase and do not form by surface diffusion. Chemical analysis performed by X-ray photoelectron spectroscopy (XPS) on the fibres confirms that no metal traces from the walls or the magnetron head can be found in the system (see Supplementary Notes 3 and 5).

**Thermodynamic mechanism for gas-phase C–C-bond formation**. Theoretical studies indicate that the stable formation of $CH_2$ radicals in the gas phase from C and $H_2$, albeit energetically favoured ($\Delta G = -3$ eV) and virtually barrierless (B = 0 eV), requires the removal of excess energy by a third body. Molecular dynamics simulations of collisions between C and $H_2$ show the formation of a metastable $CH_2^*$ intermediate. The average lifetime, where the three atoms remain bonded at distances closer than 1.9 Å, yields $CH_2^*$ lifetimes of about $\tau = 10$ ps, which agrees with previous ab initio calculations[22]. Locally, we observe temperature fluctuations due to collisions, reaching values up to $10^3$ K. The excess energy (about 3 eV) is observed to be efficiently equally shared between all the vibrational degrees of freedom immediately after the formation of the radical, preventing instant dissociation and increasing the $CH_2^*$ lifetime. Assuming that the reaction occurs in an Ar atmosphere with an estimated density greater than $10^{18}$ atoms/cm³, corresponding to our experimental conditions, collisions with a third body are expected during the lifetime of the metastable $CH_2^*$, helping to remove the excess energy and stabilising the methylene radical (see Supplementary movie 1 and supplementary Note 10). As the chain grows, more degrees of freedom are accumulated, and the lifetime increases accordingly.

In the presence of such a high third-body density, the main impediment to the growth of the hydrocarbons by continuous aggregation of $CH_2$ is the formation of ethylene

$$CH_2 + CH_2 \rightarrow C_2H_4 (\Delta G = -6.3 \text{ eV}). \quad (1)$$

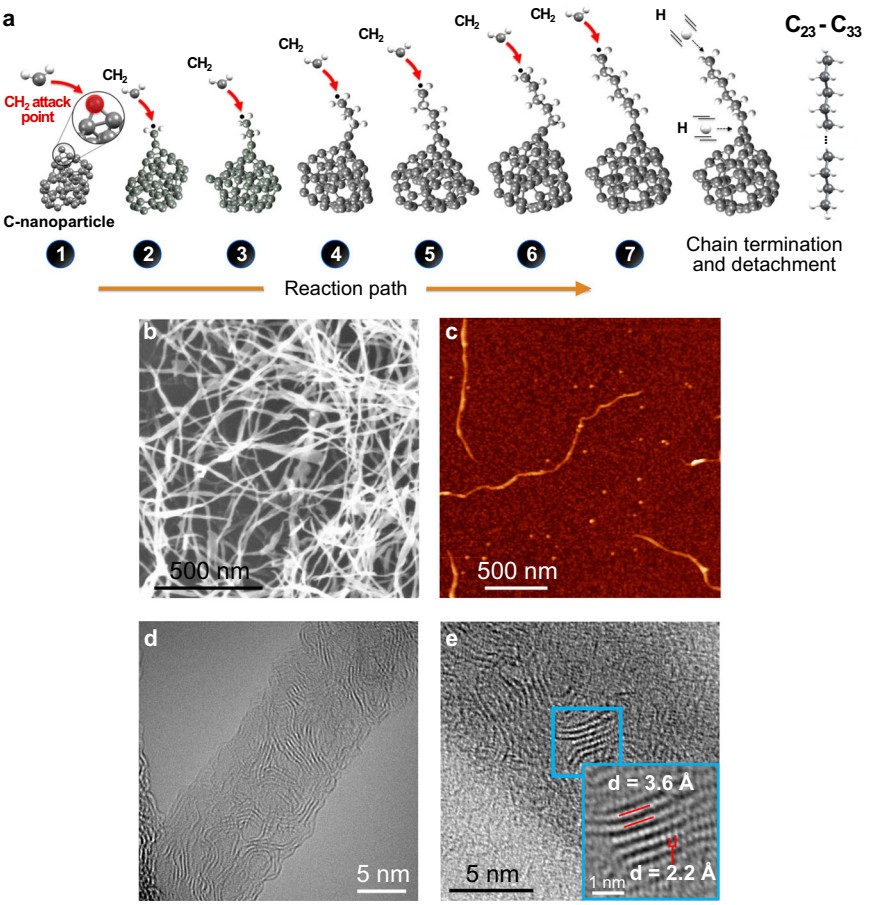

**Fig. 1 General schematic of the C–C coupling mechanism leading to long fibres. a** Pictorial views of the DFT-optimised first steps for the growth of an alkane on a model (random generated) amorphous C-NPs by sequential addition of $CH_2$ groups. **b** SEM image of a mat of fibres collected on a silicon surface. **c** ex situ AFM image of a typical individual fibre collected on a silicon surface. The bright spots correspond to amorphous C-NPs that are also produced (colour scale 0 nm, [colour bar] 30 nm). **d** $C_s$-corrected HR-TEM image of an individual fibre where locally ordered regions can be easily distinguished. **e** High-magnification image of those regions.

as the strong C=C double bond creates a barrier of about 3 eV for the addition of the next monomer

$$CH_2 + C_2H_4 \rightarrow C_3H_6 (\Delta G = -3.3 \, eV, \, B = 3 \, eV), \quad (2)$$

which is very effective in preventing hydrocarbon growth (see Supplementary Note 11). A catalyst is generally needed to avoid the aforementioned bottleneck, to assume the role of weakening the strong C=C double bond, which lowers the barrier for $CH_2$ addition and facilitates the formation of new C–C single bonds.

In our experiments, we do not take advantage of a *catalyst* as such, but the reaction

$$C - CH_2 + CH_2 \rightarrow C - C_2H_4 (\Delta G \sim -3 \, eV, \, B \sim 0 \, eV), \quad (3)$$

proceeds without a noticeable kinetic energy barrier because we exploit the fact that bonding to a contiguous C atom prevents the formation of the strong double bond in reaction (1). Such additional C atom in reaction (3) is provided in our system by a low-coordinated atom on the surface of an amorphous C-NP, also produced in the same reaction chamber. Thus, on adsorption of a $CH_2$ intermediate, the highly reactive low-coordinated atoms of the C-NPs hinder the formation of the strong double bond, inhibiting barrier formation and facilitating subsequent growth (Fig. 3a and supplementary Note 12).

The final steps are chain termination and detachment of the as-grown hydrocarbon from the C-NP surface. For both processes, atomic H is required, which can be obtained from direct dissociation of $H_2$ by the electrons generated in the sputtering

process or via chemical reactions, such as $C_2 + H_2 \rightarrow C_2H + H$[12,15,23]. The process to saturate the chain presents a negligible energy barrier, if any, and its occurrence will be proportional to the atomic hydrogen density (Fig. 3b). With regard to detachment after the alkane chain has grown on the C-NP, this can be rationalised as follows: the length of the bond between the first carbon in the alkane to the C-NPs ($d_{C1-C2}$ in Fig. 3c) expands with the length of the chain, until it saturates at 1.48 Å for 6 C units, Fig. 3b (a typical value for a single C–C bond), Fig. 3b. Therefore, this bond becomes susceptible to electrophilic attack by a H atom[24], which could arise either from diffusion on the C-NP surface or, more likely, from the gas phase. Given the barriers, we have estimated that such a process would ideally take place in the order of seconds (see Supplementary Note 13). The subsequent agglomeration and gas-phase entanglement of the detached alkane chains results in the formation of the paraffin fibres shown in Fig. 1.

**Trap and release in the flow**. As commented above, such a mechanism is thermodynamically favourable but kinetically controlled, since the sputtered C atoms are ejected at high speed from the C target and dragged by the Ar circulation. In the vicinity of the surface, there will be insufficient time for reaction with hydrogen. Furthermore, the growth of nanoparticles using sputter-gas aggregation sources is a thermodynamic none-quilibrium problemout-of-eequilibrium process[25,26]. Therefore, although our predictions indicate that growth of long

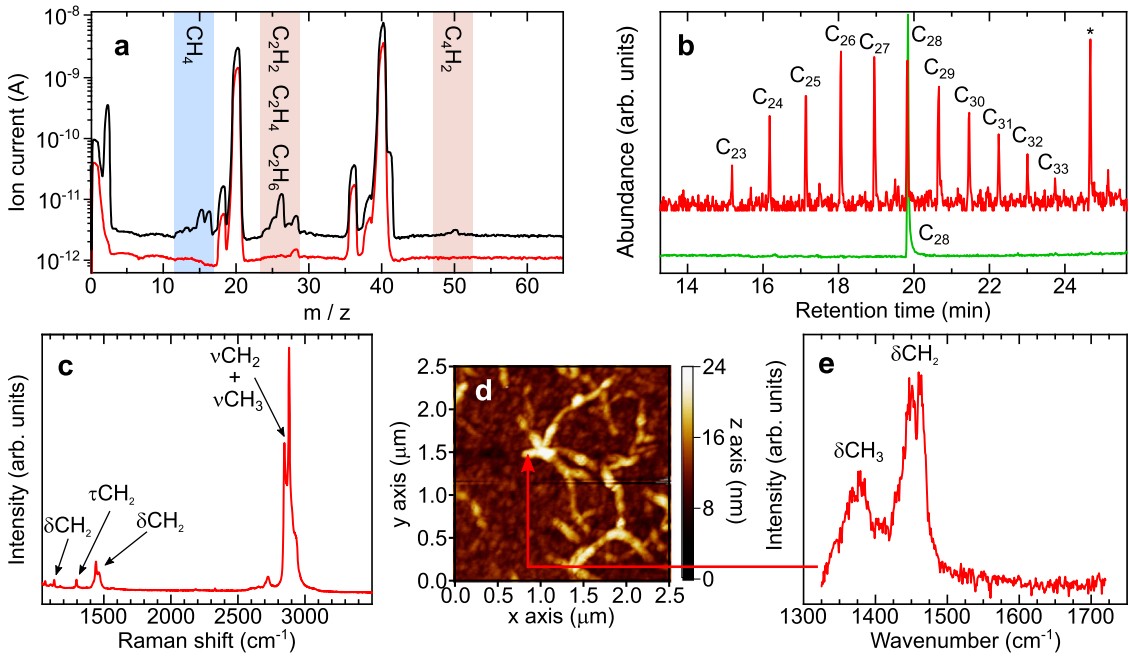

**Fig. 2 Spectroscopic composition of the fibres. a** Mass spectrum recorded with a mass quadrupole spectrometer with hydrogen introduced into the system (black line) and without hydrogen introduced into the system (red line—shifted down for clarity). The more intense unlabelled peaks, at 40 and 20 m/z, correspond to Ar (the sputtering gas) and its double ionisation, respectively, whereas the peaks at 36 and 38 m/z are due to stable Ar isotopes. **b** GC–MS chromatogram obtained from a very dilute solution of the fibres in acetonitrile collected from an Au substrate (red), showing a distribution of chain lengths centered around $C_{27}$ (peak marked * corresponds to a known line from Irganox additive from the micropipette tip). A reference spectrum of a $C_{28}$ linear alkane standard (green) was included for comparison. **c** Representative Raman spectrum recorded from a highly covered sample deposited on a $SiO_x$ substrate, with the most characteristic vibrational modes indicated. **d** AFM topography image of a fibre-rich area on an Au substrate recorded with AFM-IR (spatial resolution 10 nm) and (**e**) IR spectrum obtained at the marked position in the previous image, from a single nanofiber at 10 nm resolution, with methyl- and methylene-group vibrations indicated.

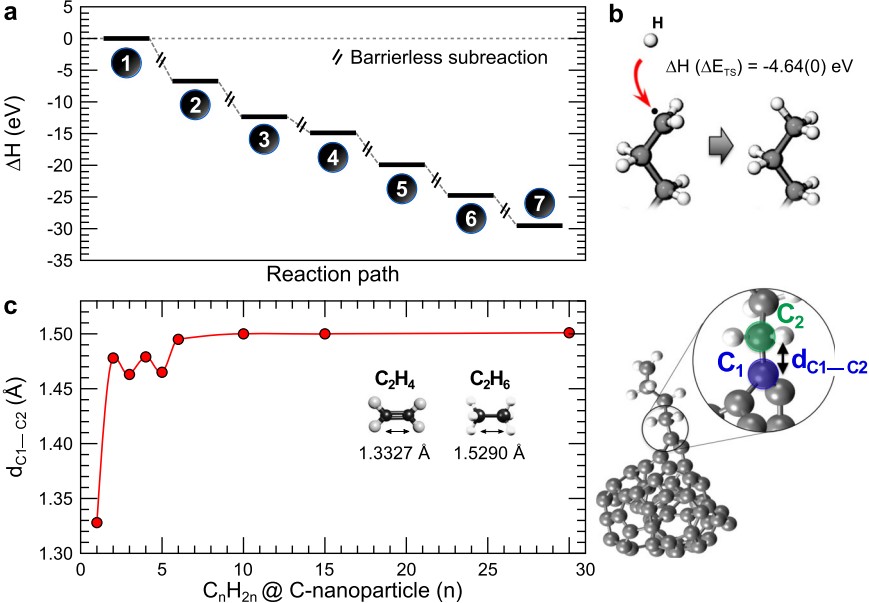

**Fig. 3 DFT energy and bond-length calculations. a** Computed enthalpy-variation diagram for each step in Fig. 1a along the reaction pathway. Enthalpy is given in eV, and all the subreactions are produced spontaneously with no thermokinetic reaction barrier. **b** Proposed mechanisms for interruption of the growth process after $C_6H_{12}$ formation on a C-NP by capture of a H to form $C_7H_{15}$, and (**c**) DFT-resulting C–C-bond length (in Å) between the carbon atoms $C_1$ and $C_2$ binding the C-NP and the growing alkane as a function of the alkane length up to $n = 30$. C–C-bond lengths for the gas-phase $C_2H_4$ and $C_2H_6$ are also shown for comparative purposes. Electrophilic H attack at the $C_1$–$C_2$ bond leads to gas-phase $C_nH_{2n+2}$.

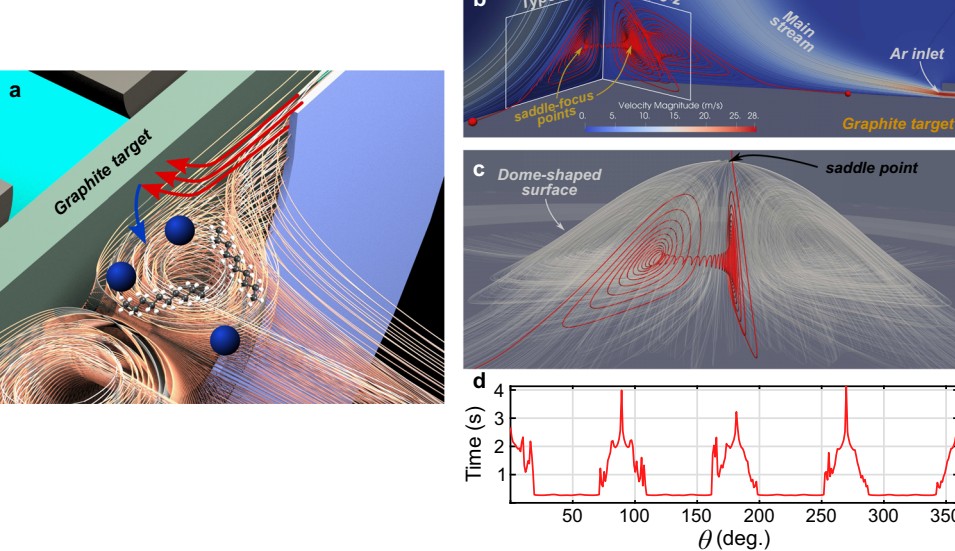

**Fig. 4 Trap-and-release flow mechanism at the magnetron head. a** General schematic of the magnetron head: orange lines represent the main Ar stream. Detail of the Ar+ ion impact (red arrows) toward the graphite target and the possible trajectory of knock out C atoms ejected toward the attraction areas (blue arrow), where gas-phase C–C coupling occurs to form C-NPs (blue spheres) and alkane chains (not at scale). **b, c** Torodial sectional streamlines ( **b**) and the stable manifold of the saddle point (**c**) in the region close to the graphite target.  Trap-and-release events for C atoms released in the flow are shown by red streamlines. **d** Residence-time distribution as a function of azimuth along the injection ring. This distribution is computed for fluid trajectories originating from a circular ring of initial conditions with a uniform grid spacing along the azimuthal and radial directions, respectively.

hydrocarbon chains is an energetically favourable process, increased residence time of the sputtered material in the region close to the magnetron is necessary[27,28]. It is clear that special kinetic factors are required to allow the reactants and products to be temporarily confined together.

Thus, the experiments were undertaken in a reaction chamber designed to increase the residence time (Fig. 4a and Supplementary Note 1). To fully understand the nature of the process, we performed direct numerical simulations of the Ar flow dynamics using our reaction-chamber geometry in order to uncover possible regions where the reacting species can be trapped in the flow and the reaction products subsequently released. From the experimental geometry, a Reynolds number Re ~20–30 and a Mach number, Ma <0.007, were estimated. At these values, unsteadiness and compressibility in the flow are not expected to play a significant role, allowing us to use a steady, incompressible simulation of the gas-phase flow. We found that these conditions give rise to a temporary trap-and-release mechanism that is already present in the steady, 3D streamline geometry of fluid particles.

Despite the toroidal recirculation zone suggested by different sectional streamline plots (Fig. 4b), the fully 3D analysis yields a solution with a fourfold broken symmetry. Specifically, a detailed tracking of judiciously chosen streamlines reveals the presence of four almost planar, invariant vertical surfaces, along with a dome-shaped surface. The vertical surfaces are 2D stable (type-1) and unstable (type-2) manifolds of saddle-focus-type points (Fig. 4b), whereas the dome-shaped surface (Fig. 4c) serves as the 2D-stable manifold of a saddle-type stagnation point[29]. Near the type-1 plane, the dome-shaped surface runs into the orifice in backward time, preventing the fluid particles under it from escaping into the main Ar jet (see Supplementary Note 14). Instead, they are forced to spiral toward the saddle-focus fixed point and then are ejected toward an adjacent type-2 plane, where the dome-shape surface does not form a barrier to the outward-spiralling particles (see Supplementary Note 14). Rather, the fluid particles

accumulate onto a 2D-separation surface, pass by the top of the dome, and enter the main Ar stream whereby they are released from the vicinity of the target. The whole process can be visualised in Supplementary Movie 2.

This interplay between type-1 and type-2 planes provides the necessary trap-and-release mechanism. In fact, arbitrarily long trapping times can arise for atoms starting arbitrarily close to type-1 planes. The expected value of the carbon atom's residence times is about 0.83 seconds (see Fig. 4d), which enables reactions that otherwise could not take place for trajectories without trapping.

This work demonstrates the efficient generation of long alkane chains entangled into fibres without the need for metal nanoparticle catalysts. Ab initio calculations reveal the role of C-NP and CH$_x$ and H radicals in the reaction pathways responsible for hydrocarbon growth. On the other hand, fluid-dynamics simulations reveal a trap-and-release mechanism demonstrating the feasibility of the process. The moderate temperatures involved (see Supplementary Note 3) and the high efficiency of the process suggest that after further development and optimisation, this methodology could lead toradically alternative routes to the clean and efficient production of valuable alkane-based chemicals. The understanding of the C–C coupling mechanisms and the dynamics of the process have a multi-disciplinary impact and offers a powerful tool for the design and optimisation of long-chain hydrocarbons based on simple and abundant building blocks.

## Methods

**sample growth**. Fabrication of carbon structures was undertaken using a scaled-up and customised multiple-ion-cluster source with three magnetrons, MICS3, from Oxford Applied Research Ltd. working in ultrahigh vacuum (UHV) (base pressure <5 × 10$^{-10}$ mbar)[11]. The schematic of the experimental magnetron head and aggregation chambers is provided in Supplementary Note 1. The 2 magnetron used for these experiments was loaded with a graphite target of 99.95% purity. The sputtering gas (Ar, 99.999% purity) was kept constant for all the experiments at 150 sccm (50 sccm injected through each of the 3 magnetrons of the MICS3). For

the experiments with hydrogen, extrapure $H_2$ (99.99% purity) was injected through the lateral entrances of the aggregation zone (named Reaction chamber in Supplementary Fig. 1) at flow rates of 1 sccm during fabrication. The typical power applied to the magnetron was 75 W. The magnetron and aggregation zone were water-cooled at 10 °C throughout the experiments.

During operation, the Reaction chamber pressure reaches about 0.1 mbar, and in the Collection chamber, $9 \times 10^{-4}$ mbar was measured. The substrates employed for collecting the nanostructures are boron-doped Si(100) with its native oxide (SiOx), polycrystalline Au, and highly oriented pyrolytic graphite (HOPG) for AFM analysis; polycrystalline Au for nano-IR; a single-crystal Au(111) surface for XPS; holey carbon TEM grids for TEM microscopy; and boron-doped Si(100) with its native oxide (SiOx) and polycrystalline Au for Raman spectroscopy. The fabricated structures were either collected in the Collection chamber (See Supplementary Fig. 1), and for some particular experiments, inside the reaction chamber.

**Scanning electron microscopy (SEM).** SEM was performed in two different instruments. SEM images of Fig. 1 and Supplementary Fig. 2 were obtained at the chemistry department of the Ben-Gurion University of the Negev, Israel. Fibres were collected on a silicon wafer, then air-dried overnight. The dried sample was sputter-coated with 5 nm. Measurements were conducted on a FEI Verios, SEM (Thermo Fisher Scientific, XHR 460 L). The images were viewed at 65,000X and 12,000X magnification, respectively, with an acceleration voltage of 3 kV. The rest of the images were taken at the Microscopy Laboratory of the Characterization Service in the Institute of Polymer Science & Technology, (ICTP-CSIC, Madrid, Spain) using a Field-Emission Hitachi SU 8000 microscope with 0.8kV accelerating voltage and 2.5 mm working distance.

**Transmission electron microscopy (TEM).** TEM observations were carried out using a cold-field emission gun in a JEOL GrandARM 300 Atomic Resolution Electron Microscope that was operated at 80 kV, conditions at which we could verify that there was no graphitisation induced by electron-beam damage to the samples. The column was fitted with a JEOL double-spherical $C_s$ aberration corrector. The fibres were directly collected in-flight by deposition onto holey TEM grids in the Collection chamber in such a way that they are self-supported, thus minimising the influence of any support.

**Mass spectrometry (MS).** MS was performed in situ in the Collection chamber with a quadrupole mass spectrometer (QMS) PrismaPlus® QMG 220 M2 (Pfeiffer) with a range of 1–100 atomic mass units. The acquisition of the data was carried out at about 30 cm after the MICS exit. Due to the pressure during operation in this chamber ($9 \times 10^{-4}$ mbar), the detection was carried out with a Faraday cup instead of using a secondary electron multiplier.

**Gas chromatography – mass spectrometry (GC–MS).** A Hewlett Packard 6890 GC gas chromatograph coupled to an Agilent Technologies model 5973 single-quadrupole mass spectrometer was employed for ex situ analysis of surface deposits. Chromatographic separation was performed on an Agilent DB-5ht poly(5% phenyl–95% methylsiloxane) fused-silica capillary column of 15 m length × 250 μm internal diameter and 0.1 μm wall thickness. The carrier gas used was helium with a flow rate of 1 mL min$^{-1}$. The sample extract was obtained by carefully washing the surface of Au slides containing fibre deposits from our experimental set-up machine, and a volume of 1 μL was injected in pulsed splitless mode at 1.5 bar during 2 min at 290 °C. The chosen chromatographic method lasted 37.5 min. The initial oven temperature was 80 °C, which was increased up to 340 °C at a controlled rate of 8 °C min$^{-1}$, then maintained at that temperature for 5 min. This chromatographic method is suitable for the optimal separation of linear hydrocarbons between around $C_{12}$ (dodecane) and $C_{49}$ (nonatetracontane). Electron impact (70 eV) was the selected ionisation type for the mass spectrometer, and the m/z = 57 identification ion was selected for determination of the n-alkanes in the sample.

**Raman spectra.** Raman spectra were recorded using a Renishaw *InVia* Reflex Raman Microspectroscopy System (Renishaw plc., Wotton-under-Edge, UK) using two different laser sources: an Ar$^{+}$ laser ($\lambda_0 = 514.5$ nm, $E = 2.41$ eV) for the samples grown in our experimental setup and a diode-pumped solid-state (DPSS) laser ($\lambda_0 = 785$ nm, $E = 1.58$ eV) for the standard n-alkanes. The laser was focused on the sample placed on the microscope stage using a 100× objective (NA = 0.85), with laser powers of <2 mW (for the 514.5 nm laser) and <20 mW (for the 785 nm laser) at the sample and a lateral spatial resolution (spot size) of between 1 and 1.5 μm. The spectral resolution was better than 4 cm$^{-1}$ and several accumulations were coadded with an acquisition time of 10 s. The spectral data were analysed using Renishaw WiRE 5.0 software. More than seventy spectra were obtained from many different grown samples with varying hydrogen-flow rates, from 0.15 to 5 sccm, and on different substrates, including SiOx and polycrystalline Au. Band-peak frequencies were obtained by curve deconvolution of the spectra using the WiRE software employing a combined Gaussian/Lorentzian line shape (20% Gaussian). The n-alkanes tricosane ($C_{23}$), tetracosane ($C_{24}$), hexacosane ($C_{26}$),

heptacosane ($C_{27}$), nonacosane ($C_{29}$), hentriacontane ($C_{31}$), and dotriacontane ($C_{32}$) were obtained from Sigma-Aldrich.

**Nano-infra-red (AFM-IR).** Nano-Infra-red (AFM-IR)[30] performed using a NanoIR2-s (Anasys Instruments) based at the SMIS-3 Beamline of Synchrotron Soleil (L'Orme des Merisiers, Paris, France) was used incorporating a tuneable mid-IR multichip MIRcat quantum cascade laser (QCL) source (Daylight Solutions). Typical experimental conditions employed were tapping frequency of the 1st mode = 395.6 kHz (detecting topography), QCL pulse rate of 356 kHz, pulse widths 300–170 ns, tapping frequency of the 2nd mode = 61.9 kHz (IR detection), and spectra scanned between 1325 and 1725 cm$^{-1}$ with a step size of 1 cm$^{-1}$. Instrument control and data collection and handling were performed using Analysis Studio v3.14.6549 software (Anasys Instruments).

In order to compare the AFM-IR spectra obtained from the fibres, Fourier transform-infra-red (FTIR) attenuated total reflectance (ATR) spectra from standard n-alkanes (detailed in the Raman section above) were recorded at a spectral resolution of 4 cm$^{-1}$ using a Specac Golden Gate horizontal ATR incorporating a heating stage in a Perkin Elmer System 2000 FT-IR equipped with a nitrogen-cooled HgCdTe detector.

**Atomic force microscopy (AFM).** AFM images were performed using the Cervantes AFM System equipped with the Dulcinea electronics from Nanotec Electronica S.L. and a Nanoscope IIIa system (Veeco) and Agilent 5500 PicoPlus (Agilent). All images were analysed using WSxM software[31].

**X-ray photoelectron spectroscopy (XPS).** XPS was carried out in situ using a PHOIBOS 100 1D electron/ion analyser with a one-dimensional delay-line detector. X-ray photons of 1486.6 eV (Al Kα emission line) were supplied by a water-cooled commercial X-ray gun XR 50 M (Specs) coupled to a Focus 500 X-ray monochromator (Specs).

**Ab initio calculations.** To benchmark different mechanisms, we have computed formation and reaction enthalpies using ab initio plane-wave density functional theory. Details of the calculations are given in Supplementary Note 9. Furthermore, transition-state (TS) geometries have been obtained by the climbing-image nudged elastic band method[32] complemented by the sequential application of the linear synchronous and quadratic synchronous transit (LST/QST) methods[33].

**Molecular dynamics.** We investigate the assembly of $CH_2$ from two-body collisions between $H_2$ and C. Alternative routes, such as collisions between CH and H, have also been explored with similar results. Such two-body reactions eventually need to dissipate their excess energy to become a viable seed eventually, e.g., through collisions with a third body, which to be effective must occur within the compound's lifetime. The experiment takes place in a dense atmosphere of Ar, which implies that collisions must occur in a timescale of the order of 10 ps.

We investigate the lifetime of the metastable compound $CH_2^*$ by ab initio-extended Lagrangian Born–Oppenheimer molecular dynamics simulations in the microcanonical ensemble. To produce realistic simulations, which at the same time could span over timescales longer than 10 ps, we utilise a plane-waves basis with on-the-fly generation (OTFG) ultrasoft pseudopotentials[34], energy cutoff of 325 eV, $10^{-6}$-eV energy threshold to reach self-consistency, and the same GGA-PBE XC functional and MP wave-function sampling[35]. Spin-polarised energy bands and a dispersion correction to include long-range van der Waals interactions have been used[36]. Finally, the reaction is simulated inside a $10 \times 10 \times 10$-Å$^3$ periodic cell box.

**Fluid-dynamics numerical calculations.** Steady-state, three-dimensional numerical simulations were performed using the open-source computational fluid-dynamics toolbox OpenFOAM[37]. The incompressible, steady-state Navier–Stokes equations were solved using the SIMPLE (Semi-Implicit Method for Pressure Linked Equations) algorithm as implemented in SimpleFoam solver[38]. Spatial derivatives are accurate to second order, and the solver is iterated, until the changes in the pressure and all three velocity components are less than $10^{-6}$ Pa and $10^{-6}$ m/s, respectively. The system was simulated for Reynolds numbers of 10, 20, and 30 based on the average inlet velocity and inlet width. Above this value, the results failed to converge due to transitioning to unsteady flow. The simulation domain was generated using the snappyHexMesh utility and has around 35-million grid cells with local refinement at the region of interest. The grid-size resolution in this region was approximately 1 μm, which is about three orders of magnitude smaller than the main dimensions on the magnetron. The inlet conditions of the simulation were uniform inlet velocity and zero normal-pressure gradient (which introduces a trivial amount of error near the inlet). The outlet conditions were uniform fixed outlet pressure and zero normal gradient of velocity (which assumes fully developed flow at the outlet). The boundary conditions on the walls were no slip for the velocity and zero-normal gradient for the pressure.

## Data availability

All data necessary to support the conclusions are present in the paper or in the Supplementary Information. All data required to interpret, verify, and extend the research are available from the corresponding authors upon reasonable request.

## Code availability

Simulation and rendering packages QUANTUM ESPRESSO, CASTEP, OpenFOAM, and Houdini Apprentice availability is guaranteed since they are free open-source codes, which can be found in https://www.quantum-espresso.org/, http://www.castep.org/, https://www.openfoam.com/, and https://www.sidefx.com/products/houdini-apprentice/, respectively.

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

## Acknowledgements

We thank Jose Criado for the artwork schematics of the system, in figures 4 and S1; Funding: We thank the European Research Council for funding support under Synergy grant ERC-2013-SyG, G.A. 610256 (NANOCOSMOS). Also, we acknowledge partial support from the Spanish MINECO through grants PID2020-113142RB-C21, PID2019-106315RB-I00, and PID2019-106315RB-I00, EU ERC CoG HyMAP 648319, and from the regional government through project S2018-NMT-4367 (FotoArt-CM). AM acknowledges to the Spanish Ministry of Science (RYC2018-024561-I), to the Regional Government of Aragon (DGA E13_20R), to the National Natural Science Foundation of China (NFSC-21850410448, NSFC- 21835002), and to the The Centre for High-resolution Electron Microscopy (ChEM), supported by SPST of ShanghaiTech University under contract No. EM02161943. S.K. and G.H. acknowledge funding from the Turbulent Superstructures Program of the German National Science Foundation (DFG)

## Author contributions

L.M., P.M., G.S., and J.A.M.-G. designed the experimental set-up and interpreted the results. L.M., P.M., G.S., and M.A. performed the growth and in situ characterisation experiments. J.I.M. and P.d.A. performed the ab initio and molecular dynamics calculations. S.K., J.A., H.S., and G.H. made the flow calculations, L.M. and L.V. performed AFM experiments, A.M. performed HR-TEM characterisation, G.S., N.S., and R.J. performed SEM characterisation, A.D., J.M. F.B, G.J.E. and J.A.M-G conducted the nano-IR experiments, V.d.P.O. contributed to the catalytic concepts and ADF analysis, G.S. and G.J.E. performed the Raman measurements on the fibres, and G.J.E. made complimentary IR and Raman measurements. E. B-B., G.J.E., R.L.-A., and J.Q.-L. made GC–MS measurements and analysed the data. J.C. contributed to the general idea of the machine design and to the choice of the reactants. P.d.A., G.H., G.J.E. coordinated the ab initio, flow calculations, and optical spectroscopy measurements, respectively. All authors discussed the results and participated in the paper preparation and editing. The whole research was coordinated by J.A.M.-G.

## Competing interests

The authors declare no competing interests.
