## [Peer Review File · Nature Communications]

Title: Metal-catalyst-free gas-phase synthesis of long-chain hydrocarbonsREVIEWER COMMENTS

Reviewer #1 (Remarks to the Author):

The paper Metal-catalyst-free synthesis of long-chain hydrocarbons via radical intermediates by Lidia Martinez et al, reports the creation of long chain hydrocarbons using a catalyst free method .The synthesis results on long fibers which are assumed to be made of unbranched Alkanes The paper is potentially very important and with industrial impact .The authors apply a tandem of characterization methods and theoretical calculations .However several points should be addressed.

- 1) The authors claim that their method is highly efficient. However, this has to be quantified and compared with catalysis-based method.
- 2) The structure of the fibers needs to be clarified. I do not think that the images are supporting the claim that they are made of non-branched Alkanes. In fact, the distances and structure appear to be turbostratic graphite This is also seeming to be supported by the AFM data. Alkanes are very sensitive to electron radiation and maybe the images correspond to a state already damage by radiation in this case the h will be kick off and the remaining carbon will rearrange as disorder graphite. Could the authors address this issue?
- 3) The section discussing the DFT calculations is a bit long and can be shorten.
- 4) The authors claim that carbon nanoparticles (most likely amorphous clusters as shown by Fig 1) are formed .However I have a hard time understanding an amorphous carbon nanoparticle .I think the C-C interaction will produce fullerenes rather than disordered clusters .Can the authors elaborate on this ?

Reviewer #2 (Remarks to the Author):

Ms Title: Metal-catalyst-free synthesis of long-chain hydrocarbons via radical intermediates

The authors have developed a novel approach to synthesize long-chain hydrocarbons. Their method is an alternative to the existing technologies especially to the Fischer-Tropsch process. Their results are clearly reported and both the experimental and theoretical parts are well conducted. The interest of their findings is undisputable and the quality of their work is of high level. Some aspects need however to be improved:

- 1) In the introduction, the text should be moderate and additional comments (most of comments are statements) are needed regarding the positioning of the reported new approach compared to the existing ones. The ecological aspect of the proposed method is only based on the used temperature and pressure and not all the cited approaches use high temperature and pressure (and non-abundant metals).
- 2) The authors have written page 5: 'In our approach, a pure gas-phase occurs...' This claim seems not in agreement with the fact that the hydrocarbon chain grow from a carbon-nanoparticle. A solid phase is required in their process. Could the author clarify this aspect?
- 3) Could the author comment on the possibility to experimentally control the hydrocarbon chain length?

What are the limitations to increase/decrease the hydrocarbon length in a controlled manner?

4) What is the exact nature of the carbon nanoparticles from which the hydrocarbon synthesis occurs?

What is the overall influence of these nanoparticles on the synthesis process: their size, crystallinity, shape etc? Could these carbon nanoparticles be introduced to the reaction chamber differently?

5) Regarding the growth mechanism, the role of CH₂ intermediates seems to be very important since it is even mentioned in the title. The revealed growth mechanism seems however based on the C-C bond nature in their process as mentioned on page 6. Could the author comment and compare their process to other existing radical C-C skeleton or network synthesis, especially regarding the nature of the radical species? And the authors are suggested to modify the title if necessary.

Reviewer #3 (Remarks to the Author):

This manuscript reports a gas phase process for hydrocarbons synthesis under mild reaction conditions using atomic carbon, molecular hydrogen and an inert carrier gas, in which the presence of CH₂ and H radical intermediates leads to efficient C-C chain growth, producing micron-length fibers of unbranched alkanes with an average length distribution between C₂₃ - C₃₃. Ab-initio calculations also uncover its potential mechanism. Such a process seems different from the conventional hydrocarbons synthesis method, however, the advantages of this synthesis process for hydrocarbons are not prominent. Some comments should be concerned.

1. Similar process has been used to prepare polycyclic aromatic hydrocarbon, which has also been reported in Nature communications (NATURE COMMUNICATIONS |(2020)11:269 | <https://doi.org/10.1038/s41467-019-14092-3> |). Based on the synthesis process, it seems to lack novelty.

2. How about the purity of the product? SEM image shows carbon particles in the product, and do they influence the purity of the final product? Moreover, how about the product yield? Please compare the purity and yield of the product between this process and the conventional synthesis process.

3. As a vital chemical, the scale-synthesis of hydrocarbons should be concerned. Such a gas phase process can prepare a small dose of the product, and is it suitable for mass production?

REVIEWER COMMENTS

Reviewer #1 (Remarks to the Author):

The paper Metal-catalyst-free synthesis of long-chain hydrocarbons via radical intermediates by Lidia Martinez et al, reports the creation of long chain hydrocarbons using a catalyst free method .The synthesis results on long fibers which are assumed to be made of unbranched Alkanes The paper is potentially very important and with industrial impact .The authors apply a tandem of characterization methods and theoretical calculations .However several points should be addressed.

1) The authors claim that their method is highly efficient. However, this has to be quantified and compared with catalysis-based method.

This is an important and difficult point that can be addressed from different points of view, in particular when a comparison with well-established catalytic methodologies is sought. For instance, Fischer-Tropsch methodologies were used in industrialized factories from 1936 and nowadays many plants have been built to produce tons of material, with continuous advances particularly in the nature of the catalysts and the architecture of the process. Any attempt to compare our new mechanism with a well-established technology would be by all means unreasonable. Indeed, this work presents a proof of concept for a new technology, which cannot be easily compared in terms of production as the system described has not been neither scaled, nor optimized for this specific chemical reaction.

Nevertheless, we have attempted to obtain some notions on the chemical efficiency of the process. Albeit, efficiency, in terms of percentage of the mass of the reactants / mass of the products, of the chemical reaction is very difficult to define due to the solid-state nature of the C-reagents, which are produced by sputtering. However, we can provide some approximate numbers considering the amount of material recovered in the fibers. We have estimated from images such as that shown in figure 1b, a density of 10^{10} fibres/cm². From this value, we can estimate a lower total deposition rate at a distance of 80 cm from the target of 10^8 fibres/cm²/minute. This number is high enough to allow for an improvement of the reaction yield after further optimization of the experimental and geometrical parameters of the system, and also suggests that the process may be susceptible to upscaling for technological production.

However, the figure of merit used in Fischer-Tropsch synthesis is the average conversion, which is typically of 45-50% reaching, in some particular cases, 90% conversion [see e.g., a paper by one of the authors: "Strong enhancement of the Fischer-Tropsch synthesis on a Co/SiO₂ catalyst activate in syngas mixture". Catalysis Communications 5 (2004) 635-638] and the control of selectivity to long chain hydrocarbons that, predicted by the ASF law, cannot exceed 45%. [see, e.g., a paper by one of the authors: "Fischer-Tropsch synthesis on mono- and bimetallic Co and Fe catalysts in fixed-bed and slurry reactors". Applied Catalysis A: General 326 (2007) 65-73]. In our case, although difficult we have estimated a conversion greater than 45% and a selectivity as high as 70% for C₂₄ +, values that are very comparable to those obtained by Fischer-Tropsch synthesis.

Moreover, although efficiency in terms of energy introduced in the system is also very difficult to assess, some hints can be also given. Our magnetron requires a power of about 75W, and the process takes place in a low-pressure environment (therefore, low gas consumption). Laboratory based Fischer-Tropsch experiments require electrical power to maintain the temperature of the system above 500K (which could be around 200W) and a significantly higher reagent consumption (high pressure).

Another way of approaching the efficiency issue is via the theoretical calculations. The process we describe can be regarded as efficient because it has very low kinetic barriers associated, as computed by our ab-initio DFT calculations.

(Further comments can be found in the answer to Reviewer 3: Question2)

Action taken: Following the reviewer's suggestion we have further discussed the efficiency including:

1. - Some sentences and two new references in the main text, page 3.
2. - A new paragraph in the supplementary information (page 4) with a specific call from the main text (page 3). This new paragraph is now entitled: Transition from nanoparticles to a full mat of fibres. Efficiency and scaling considerations.

2) The structure of the fibers needs to be clarified. I do not think that the images are supporting the claim that they are made of non-branched Alkanes. In fact, the distances and structure appear to be turbostratic graphite. This is also seeming to be supported by the AFM data. Alkanes are very sensitive to electron radiation and maybe the images correspond to a state already damage by radiation in this case the h will be kick off and the remaining carbon will rearrange as disorder graphite. Could the authors address this issue?

We agree with the referee that alkanes can be very sensitive to electron irradiation. However, the measurements were carried out with a JEOL GrandARM 300 cold field emission gun, which was operated at 80 kV to minimize beam damage of the samples. The column was fitted with a JEOL double spherical aberration corrector. The increased sensitivity of the detectors makes this kind of new generation TEMs suitable for the study of paraffins and polymeric molecules. In this respect, D. Lolla et al. [*Nanoscale*, 8 (2016) 120] studied by HR-TEM PVDF nanofibers fabricated by electrospinning and proved that this kind of microscope (operated at 80kV as in our case) allows "the acquisition of many images long before electron-induced changes to molecular scale features became apparent". Further, in this reference, polymeric chains can be clearly resolved, and they have a certain resemblance to the chain structures presented in our work.

However, for review purposes to guarantee the validity of our approach and the measurements obtained, we have repeated the electron microscopy observations at 80 kV and controlling the electron dose, see figure below. Here, we present the electron microscopy observations under very low dose conditions (the electron dose is shown in the inset) observing that no significant transformation took place and that the materials present similar structure as that shown in the main text.

Fig. Rev.1 A fibre under different irradiation doses.

Moreover, although it is true that the interatomic distances we report are similar to values for turbostratic graphite, this is also the case for alkanes or indeed other polymers, as all of them present similar interatomic values (see, for instance, references 11 and 12 of the main text).

Nevertheless, and contrary to the reviewer's appreciation, AFM does not support presence of turbostratic graphite. In the AFM images obtained, we exclusively observe rounded nanoparticles of 9 nm of diameter and fibres of around 20 nm in width and several microns in length that cannot be assigned to turbostratic graphite, which usually presents relatively flat and extended regions.

We would also like to point out that the unbranched structure of the alkanes is supported by more analytical techniques, rather than TEM. Firstly, the GC-MS data clearly supports this statement: that the main components of the nanofibres are saturated linear alkanes. The peaks in Fig. 2b were identified as linear alkanes by comparison of their retention times and mass spectra with those of linear alkane standards. The Fig. Rev.2 below shows a comparison between the data of figure 2b from the analysis of the fibres and a paraffin wax standard. As can be observed, the retention times of both samples are identical, which indicates the unbranched nature of the fibres fabricated. Moreover, this analytical technique is not subject to the sensitivity of alkanes to electronic radiation, mentioned by the reviewer. In addition to this, the Raman microscopy characterization (Figure 2 and supplementary section S6), provides definitive support the unbranched nature of the alkanes as the spectral band positions of our fibers are unequivocally assigned to *n*-alkanes by comparison to linear alkanes standards (supplementary section S6).

Fig. Rev.2. Comparison between GC-MS of the fibres, a wax standard and a pure C₂₈ sample showing closely similar retention times.

Actions taken: A clarifying remark about the absence of damage by electron irradiation has been included in the methods section.

3) The section discussing the DFT calculations is a bit long and can be shorten.

Following the referee's suggestion, we have moved most of the technical details about the ab-initio Density Functional Theory and Molecular Dynamics calculations to the Supplementary Information: cf. new sections S9 and S10.

4) The authors claim that carbon nanoparticles (most likely amorphous clusters as shown by Fig 1) are formed. However I have a hard time understanding an amorphous carbon nanoparticle. I think the C-C interaction will produce fullerenes rather than disordered clusters. Can the authors elaborate on this?

We thank the reviewer for the opportunity to explain this point. In previous work [Santoro et al., *The Astrophysical Journal*, 895 (2020) 97] we compared the resulting nanomaterial when using only C from the SGAS and after C₂H₂ injection and TEM images like that shown below are provided where the amorphous nature of the nanoparticles was clearly evidenced.

Fig. Rev.3. Image from work G. Santoro et al., *The Astrophysical Journal*, 895 (2020) 97.

While it is true that fullerenes present a more stable phase than amorphous C nanoparticles, their formation requires surpassing large energy barriers; i.e. annealing to high temperatures and later quenching, See for example Sinitsa, A. S. *et al. J. Phys. Chem. C*, 121, 24, 13396–13404 (2017), Speranza, C. *Nanomaterials*, 11(4), 967 (2021). C. Jäger's group have published many articles where they present TEM images of fullerenes, for instance *Carbon* 45 (2007) 1542–1557. This group uses laser pyrolysis to generate fullerenes where, depending on the reactor, the temperatures reached in the process can vary from 1000-1700 to 3500 K [Jäger, C. *et al. Astrophys. J. Suppl. Ser.* 166, 557–566 (2006); Jäger, C. *et al. Astrophys. J.* 696, 706–712 (2009)]. Since our experiment involves temperatures below 500K (see section S4), we can rule out formation of fullerenes. Instead, in our system we find an abundance of C nanoparticles formed in an out-of-equilibrium process.

There are many studies in the literature where fullerenes are fabricated and observed by TEM (see, for instance, the supplementary information of <https://doi.org/10.1038/nchem.482>). Again, Jäger's group have many articles in which they present TEM images of fullerenes stacked together with other graphitic nanoparticles (see for example *Carbon* 45 (2007) 1542–1557 and *Diamond & Related Materials* 18 (2009) 392–395). In all of them, the fullerenes formed present a specific structure that is different from the nanoparticles reported in this work. In particular, the particles observed in the AFM /TEM images in our work are significantly larger: a C₆₀ fullerene is around 1 nm in diameter, whereas our amorphous nanoparticles are around 10 nm.

Action taken: 1.- Some sentences explaining the nature of the C-NP have been included in the main text together with a new reference and a call to the supplementary information file.

2.-A new paragraph and a new reference have been included in the supplementary information file section S2 and S12 to justify the formation of amorphous nanoparticles.

Reviewer #2 (Remarks to the Author):

The authors have developed a novel approach to synthesize long-chain hydrocarbons. Their method is an alternative to the existing technologies especially to the Fischer-Tropsch process. Their results are clearly reported and both the experimental and theoretical parts are well conducted. The interest of their findings is undisputable and the quality of their work is of high level. Some aspects need however to be improved:

1) In the introduction, the text should be moderate and additional comments (most of comments are statements) are needed regarding the positioning of the reported new approach compared to the existing ones.

Following the reviewer's comment we have modified the introduction including some data to compare with the results presented in this paper. In particular we have focused on processes related to the objectives of our work.

The ecological aspect of the proposed method is only based on the used temperature and pressure and not all the cited approaches use high temperature and pressure (and non-abundant metals).

We agree with the reviewers comment. In the case of the use of microorganisms for hydrocarbon production the temperature and pressure are moderate and in most cases is not necessary the use of metal catalysts. However, often non-environmentally friendly or toxic organic solvents need to be employed as part of the process to obtain the alkanes. It is clear that as these methodologies improve, also the use of green and sustainable solvents (another innovation area) will be increasingly evident. The technological readiness of these processes is still relatively low, but it is a fast growing area as an alternative to less sustainable methods.

Action taken: We have improved the phrasing in the introduction section

2) The authors have written page 5: 'In our approach, a pure gas-phase occurs...' This claim seems not in agreement with the fact that the hydrocarbon chain grow from a carbon-nanoparticle. A solid phase is required in their process. Could the author clarify this aspect?

The reviewer is correct in this appreciation. With the sentence "... a pure gas-phase occurs..." we meant that the polymerization process takes place in an aerosol in a UHV chamber rather than on a surface. The resulting products are formed in the gas phase, during their flight in the gas expansion to the next chamber, but the chain growth initiates on the surface of C-NPs. However, contrary to magnetron sputtering growth techniques or Fischer-Tropsch methodologies, the hydrocarbons are not formed after a surface diffusion on the substrate or in the walls of the reactor.

Action taken: The referred to paragraph of pag. 5 has been modified to emphasize that there is no influence of surface diffusion, neither on the substrate nor on the reactor walls. We have also thoroughly examined the text and supplementary information to assure that this concept is clear, and removed the word "formed" in page 4.

3) Could the author comment on the possibility to experimentally control the hydrocarbon chain length? What are the limitations to increase/decrease the hydrocarbon length in a controlled manner?

We thank the reviewer for the opportunity to explain this point.

In our model the chain length is limited by a competition between the polymerization ($C_nH_{2n} + CH_2 \rightarrow C_{n+1}H_{2n+2}$) and the chain termination ($C_nH_{2n} + H \rightarrow C_nH_{2n+1}$) probabilities.

Polymerization requires available molecular hydrogen in order to form the CH_2 radical. However, if there is atomic H available the chain can terminate, forming a methyl moiety. As the concentration of atomic hydrogen increases, the chain termination probability will increase resulting in shorter chains.

There are two main sources of atomic H: firstly, a proportion of the molecular H_2 arrives at the magnetron head and it is dissociated. This requires that the flux of molecular hydrogen penetrates into the stream of expanding Ar. We have calculated, very approximately, that it could reach between 1-10% of the H_2 introduced. The second source is related to possible chemical reactions producing atomic H from molecular hydrogen (e.g. $CH+H_2 \rightarrow CH_2 + H$). Importantly, the production of atomic H for any of those mechanisms is proportional to the initial H_2 concentration, and therefore in any case the ratio between H_2 and H would not significantly change.

Therefore, under the current experimental arrangement there is a very narrow range of possibilities to vary this ratio and, as a consequence, we always obtain very similar chain length distributions irrespective of the H_2 flow rate employed, as we have repeatedly observed.

However, a possible method to tailor the chain length would be by changing the Ar pressure. Reduced (increased) Ar pressure, which will allow the H_2 flux to reach the magnetron surface to increase (decrease) and, therefore, the proportion of H species will also increase (decrease). Similarly, introducing H_2 /Ar mixtures using the gas entrance for the sputtering gas would lead to an important increase in the production of H and in this case, we could expect shorter hydrocarbons and also more possibilities for branched chains. However, we would like to point out that this modification may also change other factors, such as carbon erosion yield, that may induce new chemical routes, and therefore, the process will require reoptimization for this new regime. This approach has not yet been tested, because this would lead to a long optimization that exceeds the objectives of this proof of concept presented in the present manuscript.

Action taken: This discussion is a very important point, and worthy to address in detail, therefore a new discussion section in the supplementary information file has been introduced. In the supplementary section S8 devoted to a discussion of the hydrocarbon chains.

4) What is the exact nature of the carbon nanoparticles from which the hydrocarbon synthesis occurs? What is the overall influence of these nanoparticles on the synthesis process: their size, crystallinity, shape etc? Could these carbon nanoparticles be introduced to the reaction chamber differently?

The carbon nanoparticles are amorphous and previous studies (see doi: 10.3847/1538-4357/ab9086) by TEM (see fig. ref. 2) clearly indicates their amorphous nature. The reason is due the low temperatures operating during the process that leads to C aggregation in an out-of-equilibrium process. Although their size can be slightly adjustable by the experimental parameters, they present a narrow size distribution. In our experimental conditions we estimate this to be: 9 ± 2 nm

Importantly, our NP modelling performed over particles with different sizes show that the result is quite independent of the size. Although amorphous NPs of 20, 60 and 100 carbon atoms were modelled, in all cases it was observed that there is always a more prominent C atom with sp^3 hybridization that is susceptible to anchor a CH_2 and initiate the polymerization.

Unfortunately, we do not see any way of introducing the C particles differently into the experimental system, although the fact that we are producing in the same process the C NP that further acts as a starting point for alkane growth is undoubtedly attractive. However, our simulations indicate that any atom (even a molecule) capable of weakening the $CH_2=CH_2$ bond by previously forming a $C-CH_2$, could produce the same effect and,

therefore, this process could also be performed purely in gas phase by the formed $\text{CH}_2\cdot$ radical. This allows us to speculate that we could introduce any molecule susceptible to interact with $\text{CH}_2\cdot$

Action taken: 1.- Some sentences explaining the nature of the C-NP have been included in the main text together with a new reference and a call to the supplementary information file.

2.- A new paragraph and a new reference have been included in the supplementary information file section S2 and S12 to justify the formation of amorphous nanoparticles.

5) Regarding the growth mechanism, the role of CH_2 intermediates seems to be very important since it is even mentioned in the title. The revealed growth mechanism seems however based on the C-C bond nature in their process as mentioned on page 6. Could the author comment and compare their process to other existing radical C-C skeleton or network synthesis, especially regarding the nature of the radical species? And the authors are suggested to modify the title if necessary.

We thank the reviewer for the opportunity to clarify this important point. It is well-known that radical mechanisms that generate C-C bonds are the basis of many organic reactions to produce fine chemicals, polymers, pharmaceuticals, and natural products [Encyclopedia of Radicals in Chemistry, Biology and Materials; Chatgililoglu, C., Studer, A., Eds.; Wiley: New York, 2012]. In essence our process has similar aspects to conventional radical chain growth mechanism with initiation, propagation and termination steps. Indeed, one of the main differences between the diverse radical mechanisms resides in the initiation step, where several strategies can be used to produce the radicals from initiator molecules. Thermal decomposition is the most widely used, but photolysis, radiation and catalytic approaches via C-C cross-coupling, using transition metals [Metal-Catalyzed Cross-Coupling Reactions and More; de Meijere, A., Bräse, S., Oestereich, M., Eds.; Wiley-VCH: Weinheim, 2014] or small organic additives [Sun, C.-L.; Shi, Z.-J. Chem. Rev. 2014, 114, 9219–9280], are also employed.

Our case radically differs from others in that, firstly, the building blocks of atomic carbon and molecular hydrogen are the only reagents that participate in the chemical reaction, i.e. no additional initiator molecules or catalysts or a specific energy source are required for the initiation step, which is the formation of the methylene radical in the gas phase through the collision of C atoms and molecular H_2 . Secondly, the presence of the inert Ar gas provides both a third body that participates by removing excess energy $\text{CH}_2\cdot$ and generates a flow regime that increases the lifetime of the $\text{CH}_2\cdot$ radical, allowing sufficient time for gas-phase reactions that ultimately lead to chain propagation. From the mechanism described in Figure 1a, the end of the growing chain could be considered as the radical intermediate that, on collision with the highly reactive gas-phase $\text{CH}_2\cdot$ radical, leads to the formation of a C-C bond and chain propagation via subsequent steps, similar to that occurring in radical polymerization.

Chain termination takes place via combination between the hydrocarbon chain radical intermediate and a $\text{H}\cdot$ radical arising from the direct dissociation of H_2 by the electrons generated in the sputtering process or via chemical reactions, albeit we cannot ignore the much lower probability of a $\text{CH}_3\cdot$ radical termination. Note that if two radicals meet in the gas phase without the stabilising contiguous carbon initiated from the C-NP, ethene will be formed that is very stable, as explained in the text on page 6.

Action taken: Thus, we understand and agree with the reviewer's concern about the title, and since albeit the process includes concepts similar to those found in radical chain growth, it is a gas-phase mechanism far removed from conventional radical synthetic processes. Thus, we have removed the term radical

intermediates from the title and included “gas-phase” to more clearly differentiate our process from radical processes in the literature, and made some small modifications in the text to avoid further confusion. The title is now: **“Metal-free gas-phase synthesis of long-chain hydrocarbons”**.

Reviewer #3 (Remarks to the Author):

This manuscript reports a gas phase process for hydrocarbons synthesis under mild reaction conditions using atomic carbon, molecular hydrogen and an inert carrier gas, in which the presence of CH₂ and H radical intermediates leads to efficient C-C chain growth, producing micron-length fibers of unbranched alkanes with an average length distribution between C₂₃ - C₃₃. Ab-initio calculations also uncover its potential mechanism. Such a process seems different from the conventional hydrocarbons synthesis method, however, the advantages of this synthesis process for hydrocarbons are not prominent. Some comments should be concerned.

1. Similar process has been used to prepare polycyclic aromatic hydrocarbon, which has also been reported in Nature communications (NATURE COMMUNICATIONS |(2020)11:269 | <https://doi.org/10.1038/s41467-019-14092-3> |). Based on the synthesis process, it seems to lack novelty.

In reply to the reviewer we must disagree with this comment, which may be due to a misunderstanding of the message in the new mechanism we are proposing. The reference of Lemmens et al. has no relation at all with our work. Lemmens et al. make electrical discharges using a gas pulse consisting of naphthalene in argon (80 °C, 0.25%) to grow larger polyaromatic hydrocarbons (PAHs). This is totally unrelated to our case, where we aggregate C atoms and H₂ molecules to form unbranched alkanes, a high valued chemical, and we unveil the chemistry involved in a completely new mechanism. In the work suggested by the reviewer, the PAH naphthalene is the precursor to generate new PAHs by arc discharge, a methodology known to produce ions and radical species. However, in our work, simple and basic atomic building blocks react to form aliphatic chains, which currently can only be formed industrially through catalytic Fischer-Tropsch reactors, and ultimately fibre materials.

Therefore, we respectfully disagree with the reviewer's comment regarding the lack of novelty, since following the route proposed by Lemmens et al. it is impossible to obtain a controlled distribution of aliphatic compounds. We were already aware of the cited work and decided not to include it in the list of references as it is very far removed from the objectives of our research.

2. How about the purity of the product? SEM image shows carbon particles in the product, and do they influence the purity of the final product? Moreover, how about the product yield? Please compare the purity and yield of the product between this process and the conventional synthesis process.

If we consider the Fischer-Tropsch synthesis (FTS), i.e. the conversion of "syngas" (a mixture of carbon monoxide and hydrogen obtained from the gasification of natural gas, coal or biomass, as the "conventional" process), then the purity issue is highly dependent on the feedstock. Gasification is the most costly and inherently inefficient and energy demanding (typically >1000 K) and introduces many impurities that must be cleaned and conditioned before entering into the FTS reactor. The by-products of FTS (considered as impurities) are thus conditioned by the nature of the feedstock and can be very diverse including oxygen-containing hydrocarbons, light gases, etc. Other impurities arise from the catalysis type and efficiency, and the sensitivity of poisoning of the catalysts. Sulphur poisoning is an important issue in Fe-based and particularly Co-based catalysts, and a sulphur removal process must be employed. Thus, regarding yield, there are many factors to be considered (including the feedstock) and examples of carbon efficiencies (e.g. mass of C in C₅-C₂₀ range/ mass of C in feed in a high temperature FTS process) between around 23 – 41% have been reported for biomass (see Unruh, et al DOI:10.1021/ef9009185), and a conversion efficiency based on their higher heating values (H_s of C₅-C₂₀ / H_s of feedstock) of between 32 – 51% for biomass, 35 – 50%

for coal and 53-63% for natural gas, much lower than that of 94% for petroleum refining. (Additional comments in this respect can be found in the answer to Reviewer 1: Question 1)

On the contrary, in our process there are no by-products and impurities, indeed the only "impurity" arises from the presence of the C NPs along with the fibres. Their separation and elimination can be achieved in a post-processing step that should be relatively easy. Our XPS analysis, see figure S5, show that there are no traces of other contaminants. This is due to the ultra-high vacuum environment where the process takes place.

At this stage, this work provides a demonstration of the process and may be far removed from the future technology, which will of course require optimization to take into account many factors. In this respect we have added a commentary in the supplementary information that includes some broad considerations about scale-up.

Finally, the concept of yield in the context of a comparison with existing technologies is very difficult to embark upon in a rational manner. We have been able to calculate a ballpark figure of the rate of fibre production of around 2 micrograms/minute from the SEM images obtained, but by no means is the process optimized. Indeed we only collect on the surface a fraction of all of the fibres produced. Another consideration to take into account is the materials sustainability, as the Ar gas and unreacted H₂ can be recovered and reused, along with the C NPs that can be reprocessed.

Action taken: As a discussion about purity was included in the supplementary information note S5, we do not consider the need for further discussion in the text.

3. As a vital chemical, the scale-synthesis of hydrocarbons should be concerned. Such a gas phase process can prepare a small dose of the product, and is it suitable for mass production?

We agree with the reviewer in this point. This is an important issue that requires further consideration, as already mentioned for Reviewer 1. As the answer to this query is similar to the one provided to Reviewer 1 (question 1), we refer the reviewer to this point.

Action taken: Following the reviewer's suggestion we have further discussed the efficiency including a new paragraph in the supplementary information (page 4) and a specific call from the main text (page 3). This new paragraph is now entitled: Transition from nanoparticles to a full mat of fibres. Efficiency and scaling considerations.

REVIEWER COMMENTS

Reviewer #1 (Remarks to the Author):

I am still troubled by the response of the authors. For instance, in fig 1d) and e) the authors claim that it is highly crystalline. What it seems is a poly crystalline disorder graphite which appear in many cases. maybe it is compatible with Alkanes but also with many other things.

It is a pity that having so many techniques and authors they do not perform more detailed TEM. This technique is extremely important for getting the full picture.

I think the paper is incomplete and the authors claims are not fully proved.

Reviewer #2 (Remarks to the Author):

The authors have fully responded to my comments. The corrected manuscript can now be accepted for publication.

Reviewer #3 (Remarks to the Author):

The authors addressed all my concerns and comments, I have no comments on the paper.

Comments to the reviewers:

We are delighted that reviewers #2 and #3 are fully convinced with our answers and accept the paper without changes for publication. Also, we are pleased that reviewer #1 accepts three of our responses, although he/she remains uncertain with one very specific point, that we wish to clarify, as we believe it has arisen from a misunderstanding.

1.- *"in fig 1d) and e) the authors claim that it is highly crystalline"*.

We believe this is the origin of the misunderstanding: We do not claim that our fibres are highly crystalline, but instead, we indicate *"highly ordered crystalline regions that can be easily distinguished"* inside a fibre. This is a well-known and characteristic feature of semicrystalline organic materials and polymers, where ordered crystalline domains can be found amongst disordered chains, which is well documented in both textbooks and the literature [see for example; Gedde U.W., Hedenqvist M.S. (2019) Morphology of Semicrystalline Polymers. In: Fundamental Polymer Science. Graduate Texts in Physics. Springer, Cham. https://doi.org/10.1007/978-3-030-29794-7_7; Alamo R.G. (1993) The Crystallization Behavior of Long Chain N-Alkanes and Low Molecular Weight Polyethylenes. In: Dosière M. (eds) Crystallization of Polymers. NATO ASI Series (Series C: Mathematical and Physical Sciences), vol 405. Springer, Dordrecht. https://doi.org/10.1007/978-94-011-1950-4_7; Muhammad A., Turci F., Schilling T., Crystallization mechanism in melts of short n-alkane chains, J. Chem Phys (2013) 139, 214904 <http://dx.doi.org/10.1063/1.4835015>, and many others]. Indeed, TEM can effectively reveal multiscale features from semicrystalline soft matter, many exhibiting very similar features to those we observe in our fibres [see for example Lolla 2016, DOI: <https://doi.org/10.1039/C5NR01619C>; Kwei 2020, DOI: <https://doi.org/10.1016/j.mser.2019.100516>].

The Raman data concurs with the semicrystalline nature of the alkanes formed. The band identified as **g** in Table S1 and Figure S7 is a characteristic Raman active Ag mode that only appears in the Raman spectra when two adjacent methylene chains in an all-*trans* conformation lie in an orthorhombic crystal . The TEM image in Fig.1e provides very nice evidence for these small locally ordered regions that are highly ordered compared to the amorphous material surrounding them. The evidence for the semicrystalline nature of the alkanes is discussed in more detail in the supplementary information.

Action taken:

Nevertheless, we agree that the term "highly" is not appropriate and leads to confusion, and to avoid any further misunderstanding for non-specialized readers in polymers we introduce the following changes:

1.- The sentence (in Figure heading): *"C_s-corrected HR-TEM image of a fibre where the highly ordered crystalline regions can be easily distinguished. (e) High magnification image of the crystalline regions."*

It has been replaced by: *"C_s-corrected HR-TEM image of an individual fibre where locally ordered regions can be easily distinguished. (e) High magnification image of those regions."*

The corresponding action has been made for Figure S2.

2.- A call to the Supplementary Information S6 is included in the main text, where it is explained the semicrystalline nature of the fibres. A new sentence in this section has been included relating this concept with the TEM images.

3.- References 11 and 12 have been replaced by others more specific to HR-TEM imaging of polymeric materials.

4.- The experimental details regarding the TEM have been modified to clarify the absence of electron beam damage.

2.- What it seems is a poly crystalline disorder graphite which appear in many cases. maybe it is compatible with Alkanes but also with many other things.

Indeed, we agree with the reviewer's comment, and this is the precise reason why we have employed many other characterization techniques with chemical sensitivity down to the sub-picogram and nm range. In this way we can unequivocally confirm the formation of alkane chains and their growth and aggregation into fibres. We explain this evidence throughout the text, but in order to specifically underline the most important issues related to the reviewer's concerns, please consider the following points:

1.- The reviewer refers to "disordered polycrystalline graphite", as in point 2 of the reviewer's first appraisal of our work suggesting "turbostratic graphite" rather than alkanes. Although in our answer we refuted this claim based on the literature and our own comprehensive experience in this area, we wish to underline the reasons why we can completely rule out this suggestion:

- Turbostratic graphite is generally formed under conditions of high temperature (800 – 1500°C), or severe chemical processing (such as the chlorination of carbides, etc.) also at high temperature, or mechanical processing (ball milling, etc.), or via pyrolysis or carbonization of molecular (polymer) templates (once again high temperatures). These are very aggressive conditions and very different to those in our experimental setup. It is absolutely impossible to form turbostratic graphite under the experimental conditions present in our setup.
- Concerning the possibility that radiation damage in the *ex-situ* TEM measurements could produce turbostratic graphene, as suggested by the reviewer in the following comment from the first review: *"Alkanes are very sensitive to electron radiation and maybe the images correspond to a state already damage by radiation in this case the H will be kick off and the remaining carbon will rearrange as disorder graphite"*, we showed in our previous reply (new TEM experiments were performed) that no damage was inflicted under the conditions employed.
- However, if by some highly unlikely quirk turbostratic graphite was formed, then other techniques should ratify its presence, for example Raman spectroscopy. The Raman spectra of turbostratic graphite is highly characteristic, and like graphite or any other graphene-like materials, it shows D, G and 2D vibrations near 1350, 1600 and 2700 cm^{-1} , respectively. We

did observe broad bands near to 1350 and 1600 cm^{-1} in some of the spectra (see for example Figure S.9, supplementary information) that, in agreement with the prolific amount of literature available since the early 1970's to the present [examples Tuinstra & Koenig (1970) DOI: 10.1063/1.1674108; Pimienta et al (2006) DOI: 10.1039/b613962k; Lenski & Fuhrer (2011) DOI: 10.1063/1.3605545; Bukalov, *et al* (2019) DOI: 10.5185/amlett.2019.2268; Dias *et al* (2021) DOI: 10.3390/lubricants9040043], most likely arise from amorphous carbon clusters (not turbostratic graphite). Most definitively, the 2D band expected for turbostratic carbon between 2660-2700 cm^{-1} was NOT OBSERVED in any of the Raman spectra (see Fig. S.7).

2.-Using the photothermal AFM-IR technique (see Figure 2e-f, and sup.inf. S7), we have measured infrared spectra directly from various fibres (the same series of fibres as those measured by TEM and SEM). Unlike the Raman spectrum, the infrared spectrum of turbostratic graphite is featureless unless it contains substantial amounts of contaminants from the processes used to produce it. Using nano-IR spectroscopy, the spectral signature obtained directly from the fibre is highly characteristic of the presence of saturated hydrocarbons, with CH_2 and CH_3 deformation vibrations clearly observed, as shown in Figure 2d and S10 and discussed in the supplementary section (S.7), being absolutely congruent with alkanes. The spectra obtained in NO way correspond to turbostratic carbon.

3.- Finally, as explained in our previous answer: the unbranched structure of the alkanes is demonstrated by techniques more analytical than TEM. GC-MS data clearly confirm the statement that the main components of the nanofibres are saturated linear alkanes. The peaks in Fig. 2b were unequivocally identified as linear alkanes by comparing their retention times and mass spectra with those of linear alkane standards. This is an absolute determination.

3.- It is a pity that having so many techniques and authors they do not perform more detailed TEM. This technique is extremely important for getting the full picture.

The claim of our work is so surprising – we form alkanes by the aggregation of C atoms and hydrogen in the gas phase – that we have required the combination of several analytical techniques to convince ourselves of such a conclusion. We have used the following spectroscopies: Raman, *in-situ* mass-spectrometry, gas chromatography-mass spectroscopy (GC-MS), and Nano-Infrared (AFM-IR); structural techniques: SEM, HR-TEM, and AFM; and mechanical and first-principle calculations in order to understand the process well. Only with this combination have we obtained an accurate, convincing, complete picture of such complex formation mechanism. We sincerely consider that more TEM images will not throw any further light on this.

REVIEWERS' COMMENTS

Reviewer #1 (Remarks to the Author):

I disagree with the authors on the information that can be obtained by TEM .The spatially and resolved information of TEM is a great advantage over
Nevertheless the clarification of the term Highly crystalline for Highly ordered is adequate.I do not object to the publication of this paper,